# Cooperativity of catalytic and lectin-like domain of *Trypanosoma congolense* trans-sialidase modulates its catalytic activity

**Mario Waespy**[1]*, **Thaddeus Termulun Gbem**[2,3†], **Nilima Dinesh Kumar**[4], **Shanmugam Solaiyappan Mani**[1], **Jana Rosenau**[1], **Frank Dietz**[1], **Sørge Kelm**[1,2]

**1** Centre for Biomolecular Interactions Bremen, Faculty for Biology and Chemistry, University Bremen, Bremen, Germany, **2** Africa Centre of Excellence for Neglected Tropical Diseases and Forensic Biotechnology, Ahmadu Bello University, Zaria, Nigeria, Africa, **3** Department of Biology, Ahmadu Bello University, Zaria, Nigeria, Africa, **4** Department of Biomedical Sciences of Cells and Systems, University Medical Center Groningen, University of Groningen, Groningen, Netherlands

† Deceased.
* ma_wa@uni-bremen.de

**Data Availability Statement:** All relevant data are within the manuscript and its Supporting Information files.

## Abstract

Trans-sialidases (TS) represent a multi-gene family of unusual enzymes, which catalyse the transfer of terminal sialic acids (Sia) from sialoglycoconjugates to terminal galactose or *N*-acetylgalactosamine residues of oligosaccharides without the requirement of CMP-Neu5Ac, the activated Sia used by typical sialyltransferases. Enzymes comprise a N-terminal catalytic domain (CD) followed by a lectin-like domain (LD). Most work on trypanosomal TS has been done on enzymatic activities focusing on the CD of TS from *Trypanosoma cruzi* (causing Chagas disease in Latin America), subspecies of *Trypanosoma brucei*, (causing human sleeping sickness in Africa) and *Trypanosoma congolense* (causing African Animal Trypanosomosis in livestock). Previously, we demonstrated that *T. congolense* TS (TconTS)-LD binds to several carbohydrates, such as 1,4-β-mannotriose. In this study we investigated the influence of TconTS3-LD on Sia transfer efficiency of TconTS1a-CD by swapping domains. *in silico* analysis on structure models of TconTS enzymes revealed the potential of domain swaps between TconTS1a and TconTS3 without structural disruptions of the enzymes overall topologies. Recombinant domain swapped TconTS1a/TS3 showed clear Sia transfer activity, when using fetuin and lactose as Sia donor and acceptor substrates, respectively. While Sia transfer activity remained unchanged from the level of TconTS1a, hydrolytic release of free Neu5Ac as a side product was suppressed resulting in increased transfer efficiency. Presence of 1,4-β-mannotriose during TS reactions modulates enzyme activities enhancing transfer efficiency possibly due to occupation of the binding site in TconTS1a-LD. Interestingly this effect was in the same range as that observed when swapping TconTS1a-CD and TconTS3-LD. In summary, this study demonstrate the proof-of-principle for swapping CDs and LDs of TconTS and that TconTS3-LD influences enzymatic activity of TconTS1a-CD providing evidence that LDs play pivotal roles in modulating activities and biological functions of TconTS and possibly other TS.

**Funding:** The financial support of this study was provided by the Deutsche Forschungsgemeinschaft (DFG; https://www.dfg.de), project grants to SK, Ke428/8-1, Ke428/10-1), and by the Africa Centre of Excellence for Neglected Tropical Diseases and Forensic Biotechnology (ACE-NTDFB; https://acentdfb.abu.edu.ng/facilities). The funders had no role in study design, data collection and analysis, decision to publish, or preparation of the manuscript.

**Competing interests:** The authors have declared that no competing interests exist. Author Thaddeus Termulun Gbem was unable to confirm their authorship contributions. On their behalf, the corresponding author has reported their contributions to the best of their knowledge.

## Author summary

Trypanosomes are protozoan parasites causing Human African Trypanosomiasis (HAT) and Animal African Trypanosomiasis (AAT) in sub Saharan Africa. Millions of people are at risk while millions of animals die from infection causing tremendous economic losses. This has aggravated poverty on African continent. So far, there is no vaccine available against the disease. Therefore, other strategies must be developed to combat the menace. Trans-sialidases (TS), a group of structurally similar enzymes with varying enzyme activities have been implicated in the pathogenicity of the disease. We developed a strategy to modify TS to shed light towards a deeper understanding of the underlying mechanisms of these enzymes. Here we report a so far unidentified function of TS's lectin-like domain (TS-LD) and the involvement of glycans in that process. This will impact on the possible development of new strategies against the virulent factors of trypanosomiasis.

## Introduction

Trypanosomes are protozoan parasites causing trypanosomiasis in Southern America (caused by *Trypanosoma cruzi*), also known as Chagas' disease (see review by Clayton 2010 [1]), Human African Trypanosomiasis (HAT, caused by *Trypanosoma brucei ssp.*) and Animal African Trypanosomiasis (AAT, also called Nagana caused by *Trypanosoma congolense*) in livestock in Sub-Saharan Africa. AAT brings death to millions of cattle annually [2,3]. To evade insect and mammalian host immune systems, parasites developmental stage specifically express unusual enzymes termed trans-sialidases (TS). TS catalyse the transfer of terminal sialic acids (Sia) from host glycoconjugates to terminal galactose residues on target glycoproteins [4–6]. Several studies have shown that trypanosomal TS play important roles in the pathology of the disease in mammalian host [4,7–9]. Structurally, all currently known trypanosomal TS contain two domains, a N-terminal catalytic domain (CD) and a C-terminal lectin-like domain (LD), which are connected via a 23 to 25 amino acid long α-helix [6].

Whereas published studies have focused on the enzymatic activities and catalytic mechanism of TS-CDs [10–15], no experimental biological function of the LDs has been described so far.

Smith and Eichinger reported the expression and characterisation of *Trypanosoma cruzi* TS (TcruTS)/ *Trypanosoma rangeli* sialidase (TranSA) hybrid proteins, exhibiting different Sia transfer and sialidase activities [16]. They found that the C-terminal Fn3 domain (fibronectin type III), named according to its structural relation to fibronectin type III, is not only required for expression of enzymatically active TcruTS and TranSA [17–19], but also influences the overall Sia transfer and sialidase activities [16]. Amino acid sequence alignments of a well characterised sialidase from *Macromonospora viridifaciens* [20] with TcruTS revealed that R572 and E578, which are known to be essential for galactose binding of *M. viridifaciens* sialidase [20], are well conserved in TcruTS and TranSA [16]. Point mutation of one of these residues resulted in reduced sialidase activities for both enzymes and enhanced Sia transfer activity in TcruTS [16]. As a consequence of these findings Smith and Eichinger predicted both amino acid residues (R572 and E578 in TcruTS) to be involved in galactose binding of the acceptor and/or donor substrates, which would necessarily require an overall protein folding that brings the catalytic domain and the region containing the Arg and Glu of Fn3 domain (at least R and E) close together [16]. However, the resolved crystal structures of TcruTS [6] and TranSA [21,22] demonstrated that these two amino acid residues are located far away from the actual

Sia binding pocket of the catalytic domain, and therefore, they are unlikely to be directly involved in substrate binding as proposed by Smith and Eichinger [16]. Nevertheless, even if these residues do not interact with the galactose moiety of the acceptor/donor substrate, R611 and E617 of TranSA (correspond to R572 and E578 in TcruTS) were found to form an intramolecular salt bridge at the C-terminus of the LD [22] apparently indirectly influencing enzymatic activities, since its disruption was found to change Sia transfer and sialidase activities, respectively [16]. Interestingly, structural and amino acid sequence alignments of TcruTS (EMBL: AAA66352.1), *Trypanosoma brucei* (Tbru) TS (EMBL: AAG32055.1), TranSA (EMBL: AAC95493.1), *Trypanosoma vivax* (Tviv) TS (CCD20961.1) and *Trypanosoma congolense* (Tcon) TS revealed that this salt bridge is conserved among these TS, indicating a possibly essential role in trypanosomal trans-sialidase activities. Surprisingly, to the best of our knowledge no further investigations regarding functional data of the TS-LD were reported until now.

First evidence for a more pivotal role of the LDs has come from our phylogenetic analysis done separately on CD and LD of TconTS [23]. Results revealed that when aligning TconTS-LDs the two most active enzymes TconTS1 and TconTS2 grouped together. In contrast, TconTS-CDs grouped the highly active TconTS2 together with the less active enzyme TconTS3. This diversity might be the result of natural domain swapping and selection processes that have taken place in the course of evolution. In addition, we demonstrated the binding of TconTS-LD to oligomannose and oligogalactose oligosaccharides [24]. Interestingly, mannose and oligomannose oligosaccharides are not Sia acceptor substrates for the catalytic transfer [5,25,26]. However, oligomannose oligosaccharides have been found in *N*- and *O*-linked glycans on glycoproteins or as part of their glycosylphosphatidylinositol (GPI)-anchor on the parasite's surface [27–33]. Therefore, these glycans potentially function as ligand structures for TconTS-LD. Furthermore, also TS were found to be glycosylated, predominantly with *N*-linked glycans of the high-mannose type [34,35], leading to the suggestion of intermolecular interactions possibly mediated by TS-LD. Evidence for this has come from experiments demonstrating the mannose-dependent oligomerisation of recombinant high-mannosylated TconTS [24].

In this study we report a strategy and the proof-of-principle to swap CDs and LDs from different TconTS in order to investigate whether LDs modulate Sia transfer efficiency. We have previously demonstrated that TconTS enzymes exhibit different Sia transfer and hydrolytic side activity in the presence of an acceptor substrate such as lactose [23,36]. Therefore, domain swap of LDs and CDs from a highly and a less active TconTS represent a logic target model system to test the influence of LD on enzymatic activities. For these reasons we have decided to combine the CD of TconTS1a (TconTS1a-CD) with the LD of TconTS3 (TconTS3-LD) to yield domain swapped TconTS1a/TS3.

Homology model of TconTS revealed that amino acid residues localised at the contact sites between TconTS CD and LD are well conserved in the TS family. Furthermore, *in silico* data of domain-swapped TconTS1a/TS3 revealed a similar overall topology with an extensive hydrogen bond network at the interface between CD and LD. These observations support the potential of domain swap between TconTS1a and TconTS3 without disruptions of the enzymes overall structural arrangements [14].

Finally, we fused TconTS1a-CD and TconTS3-LD and consequently expressed domain swapped TconTS1a/TS3 as a model system in CHO-Lec1 cells. In order to determine the Sia transfer efficiency of TconTS1a/TS3, we quantified the transfer product 3'SL and hydrolytic side product Neu5Ac. Along this line, we were interested in how much hydrolytic side activity can be observed and thus how efficient the transfer is. This can only be investigated in the presence of a suitable acceptor substrate such as lactose.

Enzymatic activity data of TconTS1a and domain-swapped TconTS1a/TS3 demonstrated that TconTS3-LD supresses hydrolytic side reaction in TconTS1a/TS3. The existence and strength of a simultaneous and cooperative binding of CD and LD to sialic acid and oligoman-nose structures of the same substrate respectively, is decisive for enzymatic activity. Reducing or blocking the binding of TconTS1a-LD either by swapping the LD of TconTS1a with that of TconTS3 [24] or by using 1,4β-mannotriose as a competitive ligand of TconTS1a-LD during reactions increased transfer efficiency by suppressing hydrolytic side reaction. In this study we clearly demonstrate the proof-of-principle for the involvement of TconTS3-LD in modulating catalytic activities of TconTS1a-CD.

## Methods

### Materials

Unless stated otherwise, all chemicals and reagents used in this study were cell culture and ana-lytical grade. Recombinant PNGaseF endoglycosidase was from New England Biolabs, United Kingdom. *Pfu* and *Taq* DNA polymerase, *Eco*105I, *Hind*III, *Nco*I, *Not*I, *Sal*I and *Spe*I Fast Digest restriction enzymes, T4-DNA ligase, isopropyl-β-D-1-thiogalactopyranoside (IPTG), Dithiothreitol (DTT), Coomassie Brilliant Blue (Page Blue), protein molecular weight marker (PageRuler), GeneJET DNA Gel Extraction Kit, BCA Protein Assay Kit, enhanced chemilumi-nescence system (ECL-Kit), Luria Broth (LB) microbial growth medium, were from Thermo Scientific, Germany. Biozym LE Agarose was from Biozyme Scientific, Germany. StrepTactin Sepharose, purification buffers and anti-*Strep*-tag rabbit polyclonal antibody were from IBA, Germany. β-D-galactosyl-(1–4)-α-D-glucose (4α-lactose), *N*-acetyl-neuraminic acid (Neu5Ac), 3'sialyl-lactose (3'SL), β-D-glucopyranuronic acid (glucuronic acid), lyophilised Fetuin from fetal calf serum, polyethylene glycol sorbitan monolaurate (TWEEN 20), Ex-cell CD CHO media, PEI (Polyethylenimin) transfection reagent were from Sigma-Aldrich, Ger-many. Hygromycin and Gentamycin were purchased from PAA, Austria. 1–4β-D-manno-triose was from Megazyme, Ireland. Ultrafiltration units Vivacell and Vivaspin6 were from Sartorius, Germany. X-ray film were purchased from GE Healthcare, Sweden. Protino Ni-NTA Agarose and NucleoBond Midi Plasmid DNA Purification Kit were from Macherey-Nagel, Germany. Polyvinylidenedifluoride (PVDF) membranes were from Millipore, Ger-many. 96-well transparent microtitre plate were from Sarstedt, Germany. 6 mL gravity flow columns were from Biorad, Germany.

### Cloning of TconTS into modified pET28aMBP bacterial expression vector

DNA sequences encoding for TconTS1a and TconTS3 as well as their LDs were amplified from modified pDEF vector [14,23], using the corresponding set of sense and reverse primers listed in S1 Table. The resulting PCR products were subcloned into modified pET28aMBP vec-tor [24] via *Hind*III and *Bam*HI following instructions of the manufacturers.

### Introduction of Eco105I restriction site into TconTS

To insert the *Eco*105I endonuclease restriction site into the appropriate location at the hairpin loop following the domain-connecting α-helix of TconTS, sense and reverse primers were designed annealing at this target location and comprising the *Eco*105I site (S1 Table). DNA sequences coding for TconTS-CD-α-helix with the *Eco*105I site attached at the 5'-end were amplified using *Hind*III sense primer in combination with the corresponding *Eco*105I reverse primer (S1 Table). In addition, DNA sequences encoding the corresponding TconTS-LD sequence with the *Eco*105I site attached to the 3'-end were amplified using *Bam*HI reverse and

the appropriate *Eco*105I sense primer (S1 Table). Both PCR products were digested using *Eco*105I Fast Digest restriction enzyme (Thermo Scientific, Germany), purified using GeneJET DNA Gel Extraction Kit (Thermo Scientific, Germany) and blunt-end ligated using T4-DNA ligase, following instructions of the manufacturers, to generate full TconTS sequence with the *Eco*105I restriction site inserted. Appropriate TconTS DNA sequences were cloned into modified pET28aMBP expression vector using *Bam*HI and *Hind*III restriction enzymes according to manufacturers instructions. All sequences and insertions were confirmed by DNA sequencing at the Max Planck Institute for Marine Microbiology, Bremen, Germany.

## Recombination of CDs and LDs from different TconTS

To generate domain swapped TconTS1a/TS3, corresponding pET28aMBP plasmid encoding TconTS3 was digested with *Eco*105I and *Hind*III Fast Digest restriction enzymes (Thermo Scientific, Germany) to isolate the sequence encoding TconTS3-LD, which was subsequently cloned into *Eco*105I and *Hind*III digested pET28aMBP plasmid coding for TconTS1a-CD, following the manufacturers instructions.

For expression of secreted TconTS constructs in CHO-Lec1 cells, DNA sequences coding for mutated TconTS, only comprising the *Eco*105I endonuclease restriction site, and domain swapped TconTS1a/TS3 were subcloned into the modified pDEF expression vector using *Spe*I and *Hind*III restriction sites as described previously [14].

## Purification of recombinant TconTS expressed by CHO-Lec1 cells or *E. coli* Rosetta (DE3) pLacI

Recombinant TconTS constructs were expressed by CHO-Lec1 cells or *E. coli* Rosetta (DE3) pLacI and subsequently purified as described previously [14,24].

For stable transfection in CHO-Lec1 cells, DNA sequences coding for domain swapped TconTS1a/TconTS3 as well as sequences for TconTS1a* were subcloned into the mammalian expression vector pDEF. Stable transfection, single clone selection, cultivation of TconTS expressing cells as well as purification of recombinant proteins was done as described previously [14].

Recombinant TconTS constructs expressed by CHO-Lec1 cells were purified employing *Strep*-tag chromatography and characterised by SDS-PAGE and Western blot analysis as described previously [14] and subsequently quantified by BCA assay. Cells transfected with TconTS1a* and TconTS1a/TS3 respectively, produced up to 5 mg/L of secreted protein in the cell culture supernatant directly after clonal selection.

For expression of TconTS constructs by *E. coli* Rosetta (DE3) pLacI, colonies freshly transformed with pET28aMBP plasmid, encoding the TconTS construct, were used for an overnight culture in Luria Broth (LB) medium containing 50 μg/mL kanamycin, incubated at 37˚C and 240 rpm shaking. For large scale, 1 L of LB medium containing 50 μg/mL kanamycin was inoculated with 2 mL of the overnight culture and grown at 37˚C and 240 rpm until an optical density of 0.5 at 600 nm was reached. Induction was done using Isopropyl-β-D-1-thiogalactopyranoside (IPTG, 0.1 mM final concentration) for 30 min at 37˚C followed by an induction for 14 h at 4˚C and 240 rpm. *E. coli* Rosetta (DE3) pLacI cultures expressing TconTS-LDs were induced with IPTG (0.1 mM final concentration) for 2 h at 37˚C and 240 rpm. Purification of recombinant TconTS was done as described previously employing double affinity chromatography using Ni-NTA and *Strep*-tag chromatography [24]. Purified proteins were characterised by SDS-PAGE and Western blot analysis and quantified using BCA assay according to instructions of the manufacturers.

## Trans-sialidase reactions of recombinant TconTS constructs expressed by CHO-Lec1 cells and *E. coli*

Purified recombinant TconTS enzymes were assayed for Sia transfer activity and hydrolytic release of Neu5Ac as a possible side product using fetuin and lactose as Sia donor and acceptor substrates as described before [14]. In general, TconTS reactions in 50 μL reaction volume containing 10 mM phosphate buffer pH 7.4, the appropriate amount of recombinant TconTS enzyme (50 ng of TconTS expressed by CHO-Lec1 and 1 μg of enzyme expressed by *E. coli*) as well as fetuin (100 μg corresponding to 600 μM fetuin bound Sia) and lactose (2 mM final concentration) as Sia donor and acceptor substrates were incubated at 37˚C for the times indicated. To determine the influence of 1–4β-mannotriose on enzyme activities of recombinant TconTS1a expressed by CHO-Lec1 cells, 0.25 μmol (5 mM final concentration) of the trisaccharide were additionally added to the reaction mix described above. Sia tranfer product 3'SL and hydrolytic side product Neu5Ac were quantified from chromatograms obtained, employing **h**igh **p**erformance **a**nion **e**xchange **c**hromatography with **p**ulsed **a**mperometric **d**etection (HPAEC-PAD) utilising a Dionex DX600 system in combination with a CarboPac PA100 column (Dionex/ Thermo Scientific, Germany).

## SDS-PAGE and Western blot analysis

Protein samples were separated employing SDS-PAGE as described previously [37] using a MiniProtean III electrophorese Unit (Bio-Rad, Germany) and stained with Coomassie Brilliant Blue (Thermo Scientific, Germany).

Western blot analysis was performed using primary rabbit anti-*Strep*-tag and mouse anti-TconTS mAb 7/23 for detection of recombinant TconTS as previously described [14,24]. After blotting, membranes were developed using enhanced chemiluminescence system (ECL-Kit, Thermo Scientific, Germany) and X-ray film (GE Healthcare, Sweden) according to instructions of the manufacturers. X-ray films were developed in the same way as that for Western blot analysis.

## Homology modelling and *in silico* calculations

Homology models of TconTS were calculated employing the molecular modelling software YASARA 13.3.26 [38–43] as previously described [14]. In brief, crystal structure of *Trypanosoma cruzi* trans-sialidase [6] was used as a template structure (PDB: 3b69) for calculating the models. YASARA *homology modelling* module was modified manually from the default settings of the program: Modelling speed: slow, PsiBLASTs: 6, EValue Max: 0.5, Templates total: 1, Templates SameSeq: 1, OligoState: 4, alignments: 15, LoopSamples: 50, TermExtension:10. All homology models were energy minimized using the Molecular Dynamics module of YASARA with default settings. The molecular surface was calculated using the ESPPME (Electrostatic Potential by Particle Mesh Ewald) method of YASARA *Structure* with the following parameters: Force field: AMBER96 [44], Algorithm used to calculate molecular surface: numeric, Radius of water probe: 1.4 Å, Grid solution: 3, Maximum ESP: 300 kJ/mol. Structural alignment of TconTS was generated pairwise based on structure using the MUSTANG (Multiple Structural Alignment Algorithm) module of YASARA *Structure* [45].

Amino acid sequences alignments of trypanosomal TS were performed employing the *Geneious Alignment* module of the software Geneious 5.5.5, using Blosum62 Cost Matrix [46] with gap openings and extension 10 and 0.1 respectively. Adaptations and modifications were made using the same software. Increasing darkness of sheds indicates increasing number of identical amino acid residues at each position (black: 100%; dark grey: 80 to 100%; light grey 60 to 80%;

white: less then 60% similarity). Numbers on top of each sequence indicate the corresponding residue number for the appropriate trypanosomal TS sequence. Amino acid sequences of TconTS1a (CCD30508.1), TconTS2 (CCC91739.1), TconTS3 (CCD12651.1), TconTS4 (CCD12514.1), TcruTS (AAA66352.1), TbruTS (AAG32055.1), TranSA (AAC95493.1), TvivTS (CCD20961.1) were obtained from UniProt database.

## Results

### The contact sites between TconTS CD and LD

To investigate the potential of swapping CDs and LDs of TconTS we performed detailed computational analysis in order to identify common features of molecular interactions present at the interface between CD and LD. In addition, we calculated homology model of domain swapped TconTS1a/TS3 as a proof-of-principle for the recombination on an atomistic level. Using the crystal structure of *T. cruzi* TS (TcruTS) [6] as template, homology models of TconTS1-4 were calculated as described under Methods. Similar to other TS, CD and LD are localised in close proximity, connected by a 23 to 25 amino acid long α-helix (Fig 1).

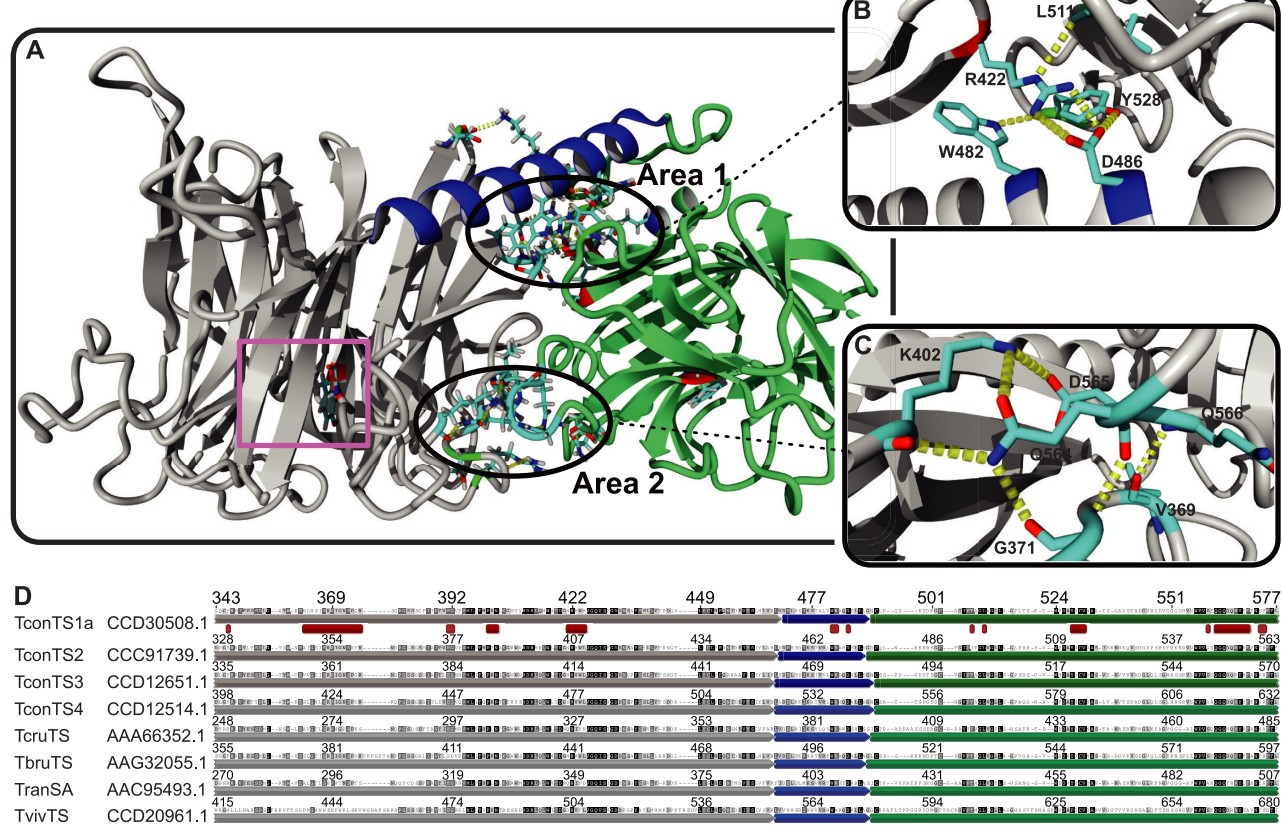

**Fig 1. Hydrogen bond network at the interface between CD and LD of TconTS. A:** homology model of TconTS1a using the crystal structure of TcruTS (PDB code: 3B69) as template structure as described under Methods. Area 1 and 2, comprising a network of hydrogen bonds formed by amino acid residues at the interface between catalytic domain (CD in grey), interdomain α-helix (blue) and lectin domain (LD in green) are marked. The active centre containing the catalytic tyrosine residue Tyr438 at the CD is labelled with a purple square **B-C:** Zoom of Area 1 and Area 2 showing the conserved hydrogen bond network (yellow dotted lines), respectively. **D:** Amino acid sequence alignment (see Methods for details) of TconTS1a through TconTS4, *T. cruzi* TS (TcruTS EMBL: AAA66352.1), *T. brucei* TS (TbruTS EMBL: AAG32055.1), *T. rangeli* sialidase (TranSA EMBL: AAC95493.1) and *T. vivax* TS (TvivTS EMBL: CCD20961.1). Only a section of the complete alignment is shown including contact sites between CD and LD. CD is labelled in grey, α-helix in blue and LD in green respectively. Sequence segments being part of the interface between CD, α-helix and LD are marked with red squares. Increasing background darkness for each residue of the sequence indicates increasing number of identical amino acid residues at the corresponding position over the alignment (see Methods for details).

**Table 1. Calculated molecular surface of the contact site between CD and LD of trypanosomal TSs and common sialidases.**

| Enzyme (PDB) | Area Å$^2$ | |
| --- | --- | --- |
| | CD/ LD | total surf. |
| TconTS1a | 1526.68/ 1500.01 | 3026.69 |
| TconTS2 | 1440.96/ 1392.59 | 2833.55 |
| TconTS3 | 1509.82/ 1331.59 | 2841.41 |
| TconTS4 | 1674.97/ 1631.50 | 3306.47 |
| TconTS1a/TS3 | 1577.43/ 1416.82 | 2994.25 |
| TcruTS (3B69) | 1483.04/ 1509.52 | 2992.56 |
| TbruTS* | 1677.44/ 1604.22 | 3281.66 |
| TranSA (1N1S) | 1470.99/ 1563.01 | 3034.00 |
| TvivTS** | 1530.39/ 1477.67 | 3008.06 |
| VCS (1W0O) | 960.88/ 935.29 | 1896.17 |
| LeechMD-SA (1SLI) | 868.53/ 918.46 | 1786.99 |

*Homology model for TbruTS was calculated using the amino acid sequence: EMBL AAG32055.1

**Homology model for TvivTS was calculated using the amino acid sequence: EMBL CCD20961.1.

From TconTS homology models, it was observed that the arrangement of CD and LD is stabilised by close contact sites between both as described for TS from other species [22]. The interface between CD and LD from TcruTS [6] and *T. rangeli* sialidase (TranSA) [22], both South American species, were determined by X-ray crystallography. Both were found to be about 36 to 48% larger compared to those from sialidases with a lectin-like domain, such as *Vibrio cholerae* sialidase (VCS) [47] and *Marcobdella* (leech) intramolecular trans-sialidase (IT-sialidase) [48]. Along this line, molecular interfaces of several African TS including TconTS1, TconTS2, TconTS3, TconTS4, *T. brucei* TS (TbruTS) and *T. vivax* TS (TvivTS) were determined from homology structure models (see Methods for details). Results revealed 33 to 46% larger areas compared to that of VCS and leech IT-sialidase for all African TS, consistent with findings for the South American species as shown in Table 1. It can be seen that the overall surface area of the contact sites between CD and LD of African TS is around 2833.55 to 3306.47 Å$^2$ and quite similar in size among the trypanosomal TS family. A detailed *in silico* analysis addressing potential interactions between amino acids at the contact sites of TconTS, revealed a network of hydrogen bonds. This network is expected to stabilise a comparative rigid overall conformation of the enzyme. Notably, calculated structure models revealed two defined locations, in which the majority of inter-domain hydrogen bonds were concentrated (Fig 1).

One site (Area 1) is closer to the α-helix connecting both domains (Fig 1A and 1B), whereas the second (Area 2) is located opposite of area 1 (Fig 1A and 1C). Amino acids of both domains forming inter-domain hydrogen bonds at the CD/LD interface of TconTS1 through TconTS4 are summarised in Table 2.

Energy minimised homology models revealed 13 hydrogen bonds formed at the interface between CD, LD and the α-helix for TconTS3, 12 for TconTS1, and 11 for both, TconTS2 and TconTS4. Interestingly, it can be seen that domain swapped TconTS1a/TS3 revealed 13 hydrogen bonds combining the interaction network of TconTS1a-CD and TconTS3-LD. Moreover, the number of these hydrogen bonds in each TS are equally distributed between both areas 1 and 2 (Fig 1B and 1C). Not surprisingly, amino acid sequence alignments of TS revealed that amino acid residues essential for hydrogen bond formation in area 1 and 2 are well conserved among the TS family (Table 3). For example, TconTS and TbruTS are highly conserved in

**Table 2. Hydrogen bonds formed by amino acid residues at the contact site between CD and LD of TconTS and domain swapped TconTS1a/TS3 as a model system.**

| TconTS1a | | | TconTS2 | | | TconTS3 | | | TconTS4 | | | TconTS1a/TS3 | | |
|---|---|---|---|---|---|---|---|---|---|---|---|---|---|---|
| Donor | Acceptor | Length (Å) | Donor | Acceptor | Length (Å) | Donor | Acceptor | Length (Å) | Donor | Acceptor | Length (Å) | Donor | Acceptor | Length (Å) |
| | | | | | | | | | | | | Y346-OH | Q559-Nε2 | 3.036 |
| | | | | | | | | | R423-Nη1 | N725-Oδ1 | 2.934 | | | |
| | | | | | | G363-N | Q557-O | 2.873 | G426-N | D620-Oδ1 | 2.885 | G371-N | Q557-O | 2.916 |
| R376-Nη2 | G563-O | 3.188 | R361-Nη1 | Q550-O | 2.889 | | | | | | | R376-Nη2 | G556-O | 2.928 |
| R376-Nη2 | S560-O | 3.141 | R361-Nε | Q550-O | 3.166 | | | | | | | | | |
| | | | R399-Nη1 | D551-Oδ2 | 2.990 | | | | | | | | | |
| | | | R399-Nη2 | D551-Oδ2 | 3.031 | | | | | | | | | |
| | | | | | | K394-Nζ | Q557-Oε1 | 2.965 | | | | K402-Nζ | Q557-Oε1 | 2.934 |
| | | | | | | K394-Nζ | D558-Oδ2 | 2.785 | K457-Nζ | D620-Oδ2 | 2.804 | | | |
| | | | | | | | | | S476-OHγ | L566-O | 2.781 | | | |
| R422-Nε | L511-O | 2.817 | | | | R414-Nε | L504-O | 2.837 | | | | | | |
| R422-Nη1 | D486-Oδ1 | 2.811 | | | | R414-Nη1 | D478-Oδ1 | 2.825 | R477-Nη1 | D541-Oδ2 | 2.892 | R422-Nη1 | D486-Oδ2 | 2.914 |
| R422-Nη2 | D486-Oδ2 | 3.019 | | | | R414-Nη2 | D478-Oδ2 | 2.854 | R477-Nη2 | D541-Oδ1 | 2.881 | R422-Nη2 | D486-Oδ1 | 2.907 |
| | | | | | | | | | | | | T478-OH | R522-Nη1 | 2.963 |
| W482-Nη1 | Y528-O | 2.928 | W467-Nε1 | Y513-O | 2.930 | W474-Nε1 | Y521-O | 2.781 | W537-Nε1 | Y583-O | 2.858 | W482-Nη1 | Y521-O | 2.838 |
| | | | K468-Nζ | E376-O | 3.346 | K475-Nζ | E383-O | 3.341 | | | | | | |
| Y528-OH | D486-Oδ1 | 2.737 | Y513-OH | D471-Oδ1 | 2.672 | T501-OH | D478-Oδ1 | 2.820 | Y583-OH | D541-Oδ1 | 2.794 | Y521-OH | D486-Oδ1 | 2.719 |
| | | | | | | G503-N | N413-Oδ1 | 2.886 | | | | | | |
| Q564-Nε2 | K402-O | 2.958 | Q550-Nε2 | E387-O | 2.877 | Q557-Nε2 | K394-O | 3.063 | | | | Q557-Nε2 | K402-O | 2.876 |
| Q564-Nε2 | G371-O | 3.182 | Q550-Nε2 | G356-O | 2.976 | Q557-Nε2 | G363-O | 2.939 | Q619-Nε2 | G426-O | 2.998 | Q557-Nε2 | G371-O | 2.933 |
| Q566-N | V369-O | 3.032 | Q552-N | V354-O | 3.115 | Q559-N | V361-O | 2.832 | Q621-N | V424-O | 2.809 | Q559-N | V369-O | 2.811 |
| N703-Nδ2 | N421-Oδ2 | 3.290 | N687-Nδ2 | G349-O | 3.371 | | | | N751-Nδ2 | G419-O | 3.020 | N696-Nδ2 | N364-O | 3.000 |

Amino acids are listed as hydrogen bond donor (Donor) and acceptor (Acceptor) being part of the catalytic domain (grey), α-helix (blue) or lectin-like domain (green).

these contact areas among each other with no more than three amino acid variations from the consensus sequence (Table 3). In contrast, TvivTS shows 10 amino acid changes relative to the consensus sequence. This is not surprising since it has been shown that TvivTS is more distant related to TconTS and TbruTS [23].

With 9 deviations from the consensus, most amino acid changes were found in TranSA, 5 in CD and 4 in LD (Table 3). Of these, 4 are identical in TcruTS, which exhibits 5 changes in total, 3 in CD and 2 in LD.

The three amino acid residues Trp serial number (sn) 10, Lys sn 11 and Asp sn 12 of the α-helix are essential for the formation of hydrogen bonds to Glu sn 6 and Arg sn 9 of CD and Tyr sn 15 and Trp sn 13 (Table 3) of LD in Area 1 (Fig 1B). Interestingly, these amino acids are conserved through all TS as listed in Table 3 except TvivTS, in which Lys sn 11 is replaced by an Asn residue, indicating a fundamental role of that region for enzyme structure preservation. In addition, it can be seen that the indole ring of Trp sn 10 provides additional van der Waals interactions with the aliphatic side chain of Arg sn 9 similar to such an interaction observed in Siglec-1 (Sialoadhesin) [49]. The Gln sn 18 of LD, which is conserved in the trypanosomal TS family, is located on a loop at the more exposed Area 2 (Fig 1C). It reaches relatively deep into the CD, where it forms hydrogen bonds to Gly sn 4, Lys sn 7 and Arg sn 5. In summary, the relatively large interface between the CD and LD of trypanosomal TS compared to related sialidases together with the extended hydrogen bond network formed by well-conserved amino acids within the interdomain interface appears to stabilise a distinct orientation of both

**Table 3. Conserved amino acid residues involved in hydrogen bond formation between CD and LD at the contact site of TS.**

| | sn | Consensus | TconTS1a | TconTS2 | TconTS3 | TconTS4 | TbruTS* | TvivTS** | TcruTS*** | TranSA**** |
|---|---|---|---|---|---|---|---|---|---|---|
| CD | 1 | Gly | N364 | G349 | G356 | G419 | G376 | G439 | G269 | G291 |
| | 2 | Arg | R368 | R353 | R360 | R423 | R380 | E443 | R273 | H295 |
| | 3 | Val | V369 | V354 | V361 | V424 | V381 | E443 | V274 | V296 |
| | 4 | Gly | G371 | G356 | G363 | G426 | G383 | W446 | G276 | T298 |
| | 5 | Arg | R376 | R361 | R368 | R431 | R388 | V451 | S281 | S303 |
| | 6 | Glu | E391 | E376 | E383 | E446 | G410 | E473 | E296 | E318 |
| | 7 | Lys | K402 | E387 | K394 | K457 | K421 | K484 | L307 | L329 |
| | 8 | Asn | N421 | N406 | N413 | S476 | N440 | D503 | Q326 | Q348 |
| | 9 | Arg | R422 | R407 | R414 | R477 | R441 | R504 | R327 | R349 |
| αHel | 10 | Trp | W482 | W467 | W474 | W537 | W501 | W569 | W386 | W408 |
| | 11 | Lys | K483 | K468 | K475 | K538 | K502 | N570 | K387 | K409 |
| | 12 | Asp | D486 | D471 | D478 | D541 | D505 | D573 | D390 | D412 |
| LD | 13 | Thr | T508 | I493 | T501 | T563 | T528 | T604 | T417 | T439 |
| | 14 | Leu | L511 | L496 | L504 | L566 | L531 | L607 | L420 | L442 |
| | 15 | Tyr | Y528 | Y513 | Y521 | Y583 | Y548 | F629 | Y437 | Y459 |
| | 16 | Ser | S560 | G546 | S553 | R615 | S580 | S663 | S468 | A490 |
| | 17 | Gly | G563 | G549 | G556 | G618 | G583 | G666 | G471 | G493 |
| | 18 | Gln | Q564 | Q550 | Q557 | Q619 | Q584 | Q667 | Q472 | Q494 |
| | 19 | Asp | D565 | D551 | D558 | D620 | D585 | D668 | N473 | T495 |
| | 20 | Gln | Q566 | Q552 | Q559 | Q621 | Q586 | R669 | Q474 | R496 |
| | 21 | Asn | N674 | N658 | N667 | N725 | H694 | A778 | D581 | D603 |

* TbruTS sequence: EMBL AAG32055.1

** TvivTS sequence: EMBL CCD20961.1

*** TcruTS sequence: EMBL AAA66352.1

**** TranSA sequence: EMBL AAC95493.1. It should be noted that different TbruTS have been identified, which group together with TconTS1, 3 and 4, respectively [23]. sn: serial number, CD: catalytic domain, αHel: α-helix, LD: lectin-like domain.

domains relative to each other. In addition, this network also seems to be formed in the homology model of domain swapped TconTS1a/TS3. Together these results provide strong evidence for the potential to combine TconTS1a-CD and TconTS3-LD resulting active TconTS1a/TS3 as a model system *in vitro*.

## TconTS domain swap

In order to investigate the influence of TconTS-LD on enzyme activities, we established a cloning strategy to swap CDs and LDs from different TconTS. As an example and model system, we combined TconTS1a-CD and TconTS3-LD and generated domain swapped TconTS1a/TS3. A structure model of TconTS1a/TS3 was calculated *in silico* and besides the conserved hydrogen bond network at the interface between CD and LD, found to predict a similar overall topology for such a recombinant TS as in the models for TconTS1a and TconTS3 (Fig 2).

 Crystal structure analysis of TcruTS [6] revealed a crystallographically unresolved flexible loop right after the α-helix between the two domains (Fig 3). This structural hairpin loop is stabilised by a well-conserved disulphide bridge formed between Cys493 and Cys503 in TconTS1a (Fig 3A). Besides these two cysteine residues, no conserved amino acids are found in this region of the four TconTS, which even differ in length (Fig 3A). In addition, in TcruTS this highly flexible hairpin loop does not play a role in the overall structure of the enzyme but

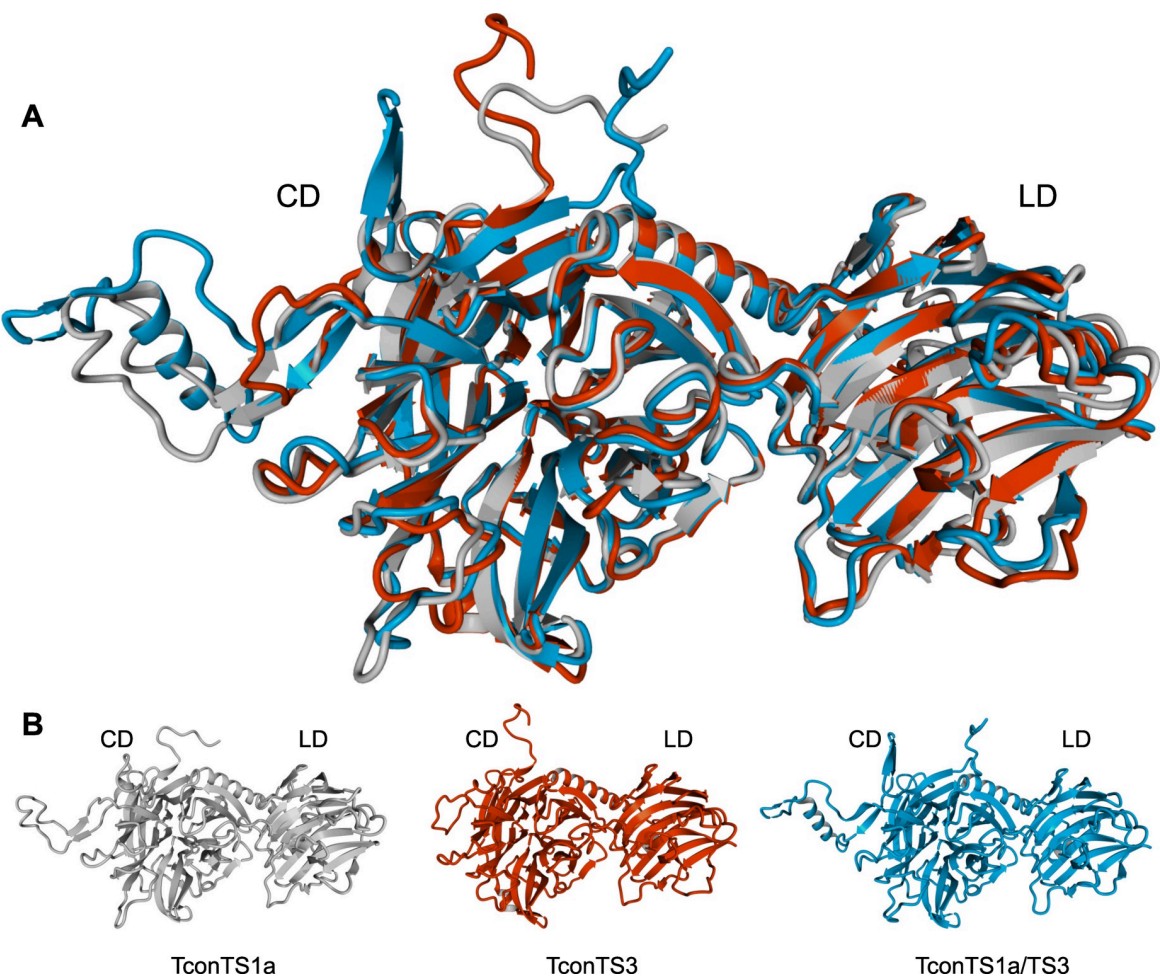

**Fig 2. Homology model of domain swapped TconTS1a/TS3. A:** Structural alignment of TconTS1a, TconTS3 and domain swapped TconTS1a/TS3 homology model. **B:** Homology model of TconTS1a (grey), TconTS3 (red) and TconTS1a/TS3 (blue). Catalytic domain (CD) and lectin-like domain (LD) are indicated. Homology models and structure alignment were generated using the YASARA *Structure* module as described under Methods.

represents a relevant antigenic region [50]. Based on these observations we hypothesised that this loop region can tolerate a wide range of mutations and thus would be suitable for the fusion of TconTS1a-CD and TconTS3-LD as well as for other domain swap constructs. Therefore, we introduced an Eco105I (SnaBI) endonuclease restriction site (coding for the dipeptide Tyr-Val) between the codes for D497 and K498 of TconTS1a and the corresponding positions in TconTS3 and the other TconTS (Fig 3B). This strategy facilitated the possibility for convenient fusion of any LD to any CD in order to swap the entire LDs, without disrupting any potentially important structure elements such as β-sheets, salt-bridges or α-helices in the enzyme.

## Expression and purification of domain swapped TconTS

In the light of previous studies [14,23,24], it has become apparent that LDs are implicated in the enzyme activity of TconTS. Therefore, domain swap of LDs from a highly and a less active TconTS represent a logic target model system. For this reason, we decided to combine TconTS1a-CD with TconTS3-LD to yield domain swapped TconTS1a/TS3 (Table 4, construct 3) in order to investigate changes in enzymatic activity relative to TconTS1a.

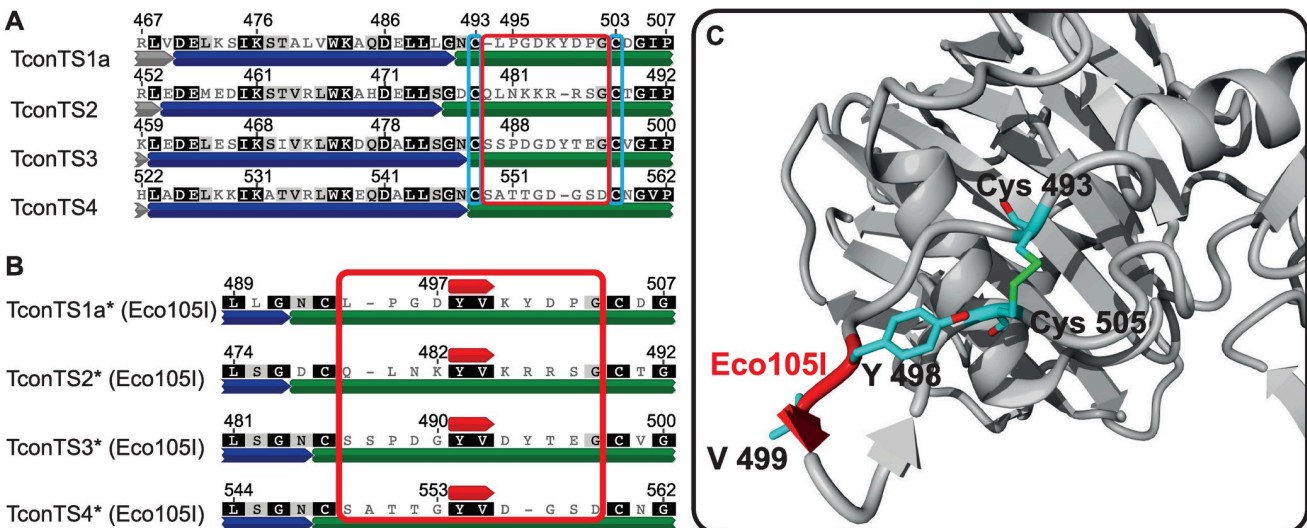

**Fig 3. Insertion of Eco105I endonuclease restriction site into TconTS. A:** A section of the complete amino acid sequence alignment of TconTS1a through TconTS4 with assigned structural elements (Grey, blue and green for CD, α-helix and LD, respectively). Cysteine residues Cys493 and Cys503 in TconTS1a forming a well conserved disulphide bridge in TS are marked with light blue frames, whereas the resulting hairpin loop flanked by these cysteine residues is marked with a red frame. **B:** Amino acid sequence alignment of the hairpin loop region with the Eco105I restriction site inserted (red annotation above each TconTS sequence, resulting in Tyr-Val insertion). Alignments were calculated as described under Methods. Numbers on top of each sequence indicate the corresponding residue numbers in the individual TconTS before the insertion. TconTS comprising the inserted Eco105I restriction site are labelled (*). Increasing darkness of background for each residue indicates increasing number of identical amino acid residues at the corresponding position. **C:** Homology model of TconTS1a* showing a region of the lectin domain (grey), including the well conserved disulphide bridge Cys493-Cys505 and the hairpin loop with the Eco105I insertion (red label). TconTS1a* homology model was calculated as described under Methods.

Recombinant TconTS1a, TconTS1a* (TconTS containing *Eco*105I restriction site, Table 4, construct 1) and domain swapped TconTS1a/TS3 were expressed in CHO-Lec1 cells as described under Methods. A schematic illustration of the expressed TconTS construct is shown in Fig 4A. Proteins were characterised by western blot analysis. When using the anti-*Strep*-tag antibody, a clear band for TconTS1a, TconTS1a* and TconTS1a/TS3 at 120 kDa respectively can be seen in the corresponding western blot (Fig 4B).

Besides the expression in CHO-Lec1 cells, we wanted to investigate whether *E. coli* could be used as an alternative expression system to efficiently yield higher quantities of active enzymes in a less time-consuming way. Therefore, recombinant TconTS1*, TconTS3* (TconTS containing *Eco*105I restriction site, Table 4 construct 2) and TconTS1a/TS3 were expressed by *E. coli* Rosetta (DE3) pLacI, as described under Methods. A schematic illustration of expressed recombinant TconTS, is shown in S1A Fig. Proteins were characterised by SDS-PAGE analysis revealing molecular masses of about 135 kDa (S1B Fig). However, only relatively low amounts (100–300 μg of enzyme per litre of bacterial culture) of soluble TconTS were obtained, whereas the majority of the recombinant proteins were insoluble. Several expression optimisations,

**Table 4. TconTS constructs generated in this study.**

| Construct number | Catalytic domain | Lectin-like domain | Swap construct |
|---|---|---|---|
| 1 | TconTS1a | TconTS1a | TconTS1a* |
| 2 | TconTS3 | TconTS3 | TconTS3* |
| 3 | TconTS1a | TconTS3 | TconTS1a/TS3 |

* Mutated TconTS containing the inserted *Eco*105I restriction site

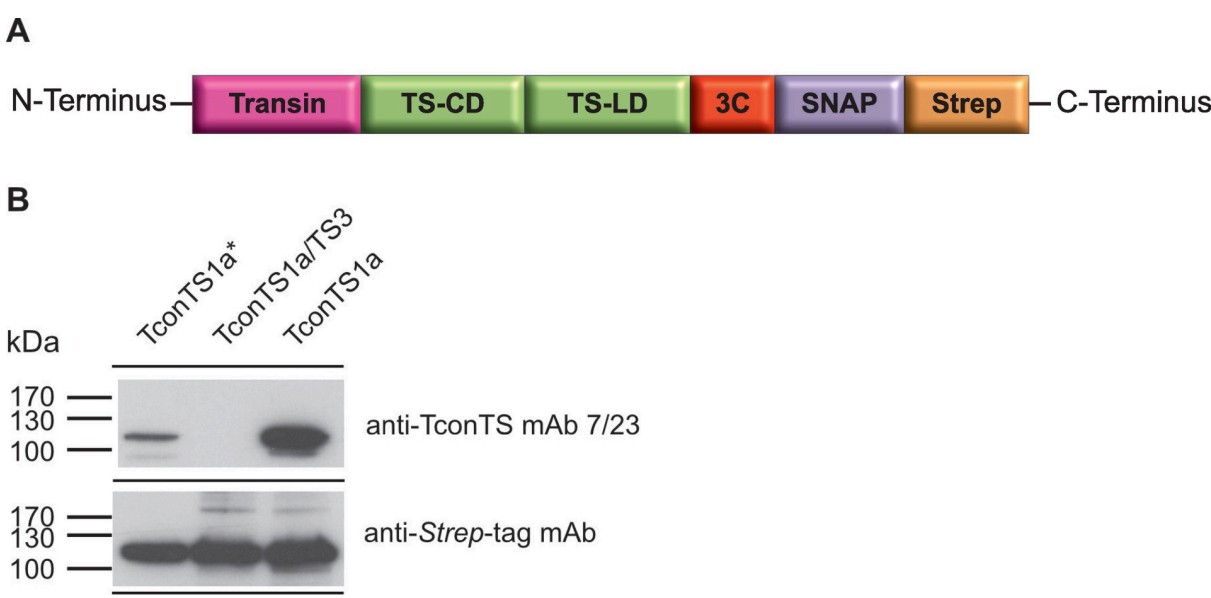

**Fig 4. Expression of TconTS constructs in CHO-Lec1 cells. A:** Schematic presentation of recombinant TconTS constructs for expression in CHO-Lec1 cells. Fusion tags flanking TconTS are: Transin: translocation signal peptide, 3C: human rhinovirus 3C protease cleavage site, SNAP: SNAP-tag, *Strep*: Strep-tag. **B:** Western blot analysis of TconTS constructs. 100 ng of each TconTS construct was used and detection was done employing anti-*Strep*-tag mAb and anti-TconTS mAb 7/23 as indicated (described under Methods).

including variation of the isopropyl-β-D-1-thiogalactopyranoside (IPTG) concentration, as well as time of induction and temperature adjustments slightly increased yields to about 200–500 µg of purified soluble protein per litre of bacterial culture.

## Binding epitope determination of the monoclonal anti-TconTS mAb 7/23

In our previous study we have shown that the monoclonal anti-TconTS antibody (anti-TconTS mAb 7/23) prepared by Tiralongo *et al.* [26] does specifically recognise recombinant TconTS1 but not TconTS3 [23]. This specificity of the anti-TconTS mAb 7/23 can be a helpful tool in order to characterise domain swapped recombinant TconTS1a/TS3. The resulting question was whether the binding epitope of the antibody is located within the CD or LD.

To answer this question, different truncated recombinant TconTS1a fragments were designed (Fig 5A). In total, five TconTS1-CD (Fig 5A constructs 1–5) and two TconTS1-LD (Fig 5A constructs 6 and 7) fragments, varying in length and containing a C-terminal *Strep*-tag, were cloned and expressed by *E. coli* Rosetta pLacI as described under Methods. Bacterial lysates containing truncated recombinant TconTS fragments were used for epitope mapping, employing Western blot analysis using anti-*Strep*-tag antibody recognising the C-terminus and anti-TconTS mAb 7/23 (Fig 5B–5D).

For all of the bacterial expressed constructs bands of similar intensities were detected by the anti-*Strep*-tag antibody at the expected molecular masses indicating that all recombinant proteins were synthesised by the bacteria at comparable levels. Notably, both TconTS51a-LD constructs 6 and 7, were also detected by the anti-TconTS mAb 7/23 (Fig 5C) indicating that the binding epitope for mAb 7/23 is in the LD. In contrast, none of the truncated recombinant TconTS1-CD constructs (1–5) were recognised by the anti-TconTS mAb 7/23 (Fig 5B). Furthermore, when testing the other recombinant TconTS2, TconTS3 and TconTS4 as well as their LDs (Fig 5D), none of them was recognised by the anti-TconTS mAb 7/23.

Interestingly, bacterial expressed TconT1a (Fig 5B lane 7) showed significantly lower signal intensity relative to TconTS1a expressed by CHO-Lec1 cells (Fig 5B lane 9). One of the major

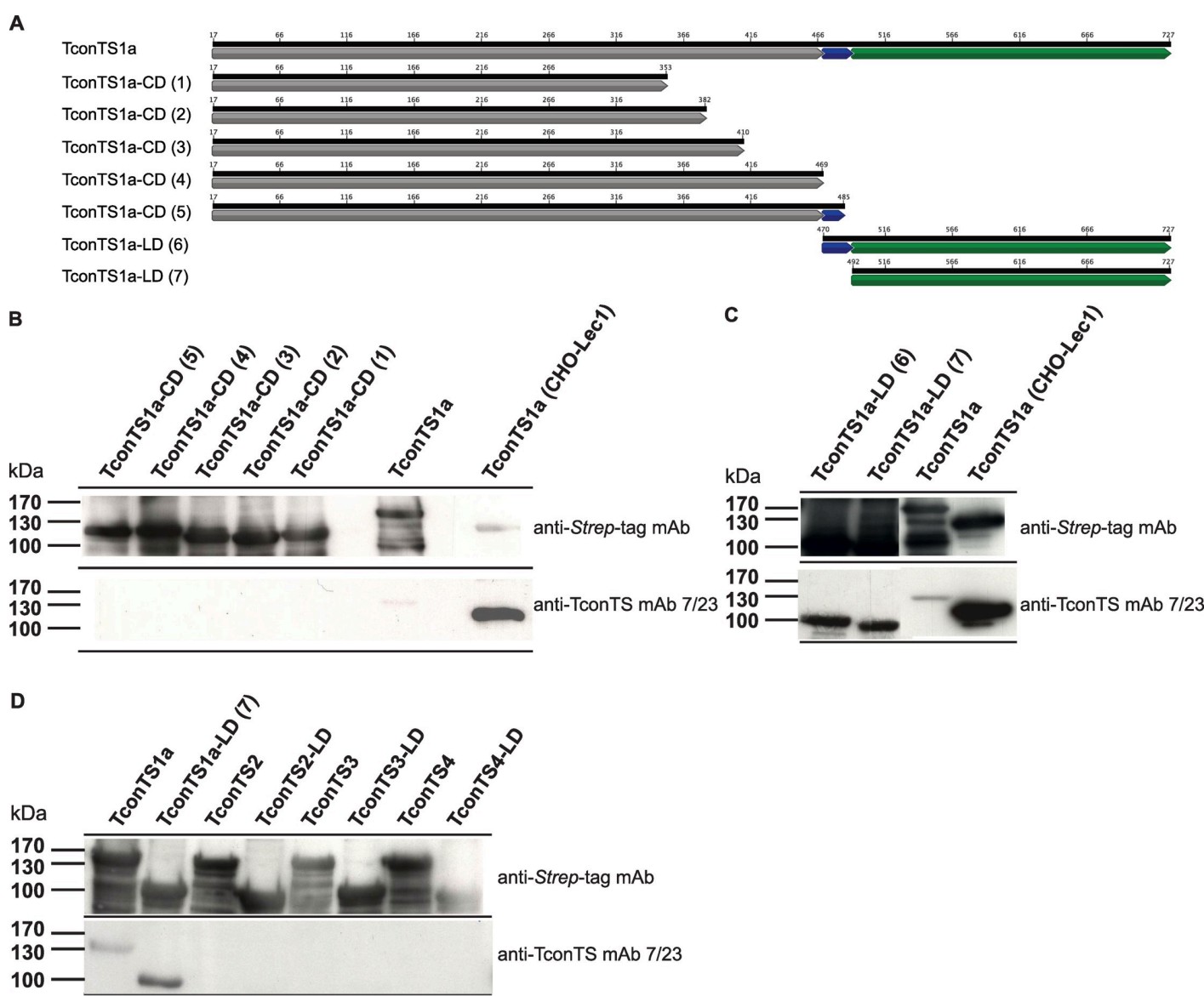

**Fig 5. Epitope mapping of anti-TconTS mAb 7/23 binding epitope. A:** Schematic presentation of wild type and truncated TconTS1a constructs (1–5, CD: catalytic domain, 6–7 LD: lectin domain) used for anti-TconTS mAb 7/23 epitope mapping. Structural elements, such as catalytic domain (CD), α-helix and lectin domain (LD) are labelled in grey, blue and green respectively. **B-D:** Western blot analysis of TconTS1a constructs (1–7) using anti-TconTS mAb 7/23 and anti-*Strep*-tag mAb as indicated (details under Methods). All TconTS1a constructs were expressed in *E. coli* Rosetta pLacI and bacterial lysates were used for SDS polyacrylamide gel electrophorese. 100 ng TconTS1a expressed and purified from CHO-Lec1 cells was used as a control in Western blot experiments.

differences between proteins expressed in eukaryotic cells and those in prokaryotic cells is the lack of *N*-glycosylation in bacteria. To investigate the possible influence on anti-TconTS mAb 7/23 antibody binding to TconTS1a, CHO-Lec1 expressed purified TconTS1a was treated with peptide-*N*-glycosidase F (PNGase F), which specifically removes *N*-glycans.

Indeed, it was found that *N*-deglycosylation of TconTS1a under native conditions drastically reduced the binding of anti-TconTS mAb 7/23 to the enzyme, indicated by the relatively weak bands at 110 kDa (S2 Fig). Furthermore, PNGaseF treatment under denaturing conditions completely eliminated the binding of anti-TconTS mAb 7/23 to TconTS1a (S2 Fig). These observations indicate that *N*-glycan structures of TconTS1a have an influence on the recognition by the anti-TconTS mAb 7/23 antibody.

**Table 5. Quantification of transfer product 3'SL and hydrolytic side product Neu5Ac using different TconTS enzymes expressed by CHO-Lec1 cells also in the presence and absence of 5 mM 1,4β-mannotriose.**

| Enzyme | Trans-sialidase activity Amount 3'SL (pmol/min/ng TS) | | Hydrolytic release of free Neu5Ac Amount Neu5Ac (pmol/min/ng TS) | Transfer efficiency |
|---|---|---|---|---|
| TconTS1a | $2.73 \pm 0.03$ | | $8.80 \times 10^{-2} \pm 6.0 \times 10^{-4}$ | 31 |
| TconTS1a* | $1.03 \pm 0.05$ | | $4.93 \times 10^{-2} \pm 7.8 \times 10^{-3}$ | 21 |
| TconTS1a/TS3 | $2.41 \pm 0.12$ | | $1.57 \times 10^{-2} \pm 2.6 \times 10^{-3}$ | 153 |
| TconTS3** | $2.61 \times 10^{-3} \pm 1.3 \times 10^{-4}$ | | n.d. (<detection limit) | >2.3 |
| **Effect of 1,4β-mannotriose***** | | | | |
| TconTS1a (1,4β-mannotriose***) | - | $2.21 \pm 0.15$ | $9.12 \times 10^{-2} \pm 6.0 \times 10^{-4}$ | 24 |
| | + | $2.16 \pm 0.21$ | $1.51 \times 10^{-2} \pm 1.5 \times 10^{-3}$ | 143 |

Quantifications of reaction products 3'SL and released free Neu5Ac were done employing HPAEC-PAD analysis as described under Methods. Transfer efficiency is defined as the ratio of 3'SL over Neu5Ac in the presence of the acceptor substrate lactose. Data points are means of triplicates ± standard deviation.

*Mutated TconTS containing the inserted *Eco*105I endonuclease restriction site

**TconTS3 kinetic data from Gbem *et al.* 2013 [23,36]. n.d.: not detected.

***TS reactions under standard conditions using TconTS1a were set up in the presence or absence of 5 mM 1,4β-mannotriose. For each reaction 50 ng of enzyme were incubated with 100 μg fetuin (600 μM fetuin-bound Sia) and 2 mM lactose for 30 min at 37°C as described under Methods.

## Activity of domain swapped TconTS1a/TS3

The crucial question of this study was which impact replacing the LD of TconTS1a with that from TconTS3 would have on the Sia transfer efficiency of the fusion protein TconTS1a/TS3.

In order to address this question, besides the production of 3'SL we wanted to investigate how much hydrolytic side activity can be detected for TconTS1a/TS3 relative to TconTS1a and thus how efficient the Sia transfer to lactose is? However, transfer efficiency can only be measured in the presence of a suitable acceptor substrate such as lactose. Therefore, we used TconTS1a/TS3 and quantified the transfer product 3'SL and the hydrolytic side product Neu5Ac in the presence of the acceptor substrate lactose.

However, it should be noted here that when using the bacterial expressed TconTS constructs in our experiments, we detected enzyme activities that were 3–4 orders of magnitude lower (S3 Fig and S2 Table) than those observed for enzymes expressed by CHO-Lec1 cells. The relatively high amount of insoluble protein indicated that the majority of TconTS is not properly folded in *E. coli* Rosetta pLacI.

Enzyme activity of CHO-Lec1 expressed TconTS was investigated using 50 ng of TconTS1a/TS3 as well as TconTS1a and TconTS1a* respectively. Enzymes were incubated with 100 μg fetuin (600 μM fetuin-bound Sia) and 2 mL lactose at 37°C for 30 min as described under Methods. Results are summarised in Table 5. TconTS1a produced about $2.73 \pm 0.03$ pmol 3'SL and $8.80 \times 10^{-2} \pm 6.0 \times 10^{-4}$ pmol Neu5Ac. In contrast, domain swapped TconTS1a/TS3 produced comparable amounts $2.41 \pm 0.12$ pmol of 3'SL but only $1.57 \times 10^{-2} \pm 2.6 \times 10^{-3}$ pmol of Neu5Ac were detected. In that respect, it can be seen that the Sia transfer efficiency has increased in case of TconTS1a/TS3 relative to TconTS1a (Table 5). TconTS1a* produced $1.03 \pm 0.05$ pmol 3'SL and released $4.93 \times 10^{-2} \pm 7.8 \times 10^{-3}$ pmol Neu5Ac respectively. Both values are lower than that of TconTS1a. However, the Sia transfer efficiency is similar for both enzymes.

We have previously demonstrated that TconTS1a-LD is a carbohydrate binding domain with a specific affinity for oligomannose oligosaccharides [24]. Hence it was hypothesised that the LD of TconTS may bind to the same substrate glycan as its CD, but at a different position. This would provide a mechanism how interaction of the LD could directly influence the over-all enzyme activity. To investigate this hypothesis, CHO-Lec1 expressed TconTS1a was incubated with 100 μg fetuin (600 μM fetuin-bound Sia) and 2 mM lactose as Sia donor and acceptor substrates under standard conditions, in the presence and absence of 5 mM 1,4β-mannotriose. Transfer product 3'SL and hydrolytic side product Neu5Ac were analysed by HPAC-PAD (described under Methods). The presence of 1,4β-mannotriose did not have a significant effect on the production of 3'SL (Table 5). Interestingly, in the presence of 1,4β-mannotriose the hydrolytic release of free Neu5Ac was about 6-fold lower than without this trisaccharide, resulting in a corresponding increase in the transfer efficiency (Table 5). It is important to point out here that this effect of 1,4β-mannotriose is in a similar range to that obtained for the domain swapped TconTS1a/TS3 relative to TconTS1a, where the increase in transfer efficiency is about factor 5 (Table 5).

## Discussion

Based on our findings, namely identifying TconTS-LD as a carbohydrate-binding domain with specific affinities to oligomannosyl oligosaccharides [24], we have hypothesised that the LD modulates TconTS catalytic activities, due to additional binding of TconTS-LD to the same substrate, influencing the overall binding affinities or catalytic turnover. To investigate the hypothesis of a cooperative interaction of TconTS-CD and LD, we established a strategy for a modular recombination of TconTS-CD and LD allowing us to efficiently swap CD and LD between different TconTS. As a proof-of-principle, we decided to fuse the CD of the highly active TconTS1a to the LD of TconTS3, a comparatively less active TS [23] resulting the domain swapped TconTS1a/TS3 as a model system to investigate cooperativity between CD and LD. Finally, we determined the Sia transfer efficiency for TconTS1a/TS3 by quantifying the transfer product 3'SL and the hydrolytic side product Neu5Ac in the presence of the acceptor substrate lactose.

Furthermore, in our previous study we demonstrated that TconTS1a-LD binds to 1,4β-mannotriose, whereas TconTS3-LD did not bind to this oligosaccharide [24]. Therefore, we investigated the impact of 1,4β-mannotriose on Sia transfer efficiency of TconTS1a. Notably, in the presence of this trisaccharide the transfer efficiency of TconTS1a was increased resembling that of the domain swap TconTS1a/TS3. These results can be explained by cooperativity between CD and LD in TconTS1a, which catalytic activity is modulated by oligosaccharide binding to its LD as discussed below.

### *In silico* structural insights into the contact site between CD and LD of trypanosomal TS reveals high level of molecular consistency

Buschiazzo and co-worker first observed that the interface between TcruTS-CD and LD is significantly larger (2550 Å$^2$) compared to other bacterial and viral sialidases (1300–1600 Å$^2$) [6]. Furthermore, Amaya and co-worker found that also the molecular surface of the contact site in TranSA is more extended (about 2600 Å$^2$) [22]. Along this line, we calculated the area of the molecular interfaces from our *in silico* models of TconTS1a, TconTS2, TconTS3, TconTS4 and domain swap TconTS1a/TS3 and observed area sizes analogous to those obtained for TcruTS and TranSA [22] (Table 1). In conclusion, in both publications [6,22] the authors predicted a relatively rigid overall TS core structure precluding a direct involvement of the LD in enzymatic catalysis. This is in agreement with our hypothesis that the LD is indirectly involved in

enzyme activity instead, potentially by modulating the affinities of TS for several donor/acceptor substrates [24]. In conclusion, the extended interfaces between CD and LD seems to be a typical feature of trypanosomal TS.

Therefore, we investigated the structural architecture at the contact sites between TconTS CD and LD in detail. Interestingly, our TconTS homology models revealed that the majority of amino acids localised at the contact site between CD and LD are well conserved among TconTS family members (Fig 1D and Table 3), indicating a high evolutionary pressure to maintain these critical amino acids in all TconTS [23]. When calculating the hydrogen bond network formed between residues at the interface between CD and LD, 11 to 13 potential hydrogen bonds were observed for TconTS1a through TconTS4, which is similar to the number found in TranSA [22]. This is in contrast to other sialidases, where only about half of the number have been found. Interestingly, domain swapped TconTS1a/TS3 revealed 13 hydrogen bonds as a combination of those observed for TconTS1a and TconTS3 respectively. Mainly two separate areas, 1 and 2, comprise the majority of hydrogen bonds formed, evenly distributed over both (Fig 1A). Area 1 is located at the region where CD, LD and the α-helix are in close contact (Fig 1A and 1B), whereas Area 2 is located more closely to the active site of CD (Fig 1A and 1C). We assume that the contacts at Area 2 keep the catalytic grove of CD (Fig 1A purple square) close to the hypothesised binding site of the LD [24], thus providing a rigid overall structure of both domains relative to each other. Strikingly, amino acid residues involved in the hydrogen bond network formation at both sites, 1 and 2, are well conserved (Table 3). Since all these amino acid residues are highly conserved among TS family despite their different catalytic activities (e.g. of TranSA), it appears unlikely that their conservation is essential for a specific catalytic activity. Nevertheless, the proposed cooperative binding of CD and LD requires a stabilised conformation of the two domains as provided by the conserved hydrogen bond networks.

## Fusion of TconTS1a-CD and TconTS3-LD results TconTS1a/TS3 exhibiting TS and hydrolytic activities

Based on the fact that the majority of amino acid residues localised at the contact sites between CD and LD are well conserved in the TconTS family, we rationalised that it might be possible to swap the domains of different TconTS to investigate the influence of LD on enzyme activities. To determine *in silico* structural stability of domain swap TconTS a structure homology model was calculated, using TconTS1a/TS3 amino acid sequence (Fig 2). In this model TconTS1a-CD and TconTS3-LD exhibited similar structure topology to the corresponding holoenzymes, TconTS1a and TconTS3 (Fig 2). Interestingly, when investigating the interface between CD and LD of TconTS1a/TS3, 13 hydrogen bonds were observed formed by amino acid residues, which were predicted to be essential for conformation stability of the wild type TconTS as discussed above. These observations underline the possibility that such an rearrangement of TconTS genes has occurred during evolution, which would provide an explanation for the different phylogenetic relationships of CD and LD in TS from African trypanosomes [23].

At first sight, the α-helix sequentially located between CD and LD itself might present a potential target for the restriction site introduction, but amino acid sequence alignments of TconTS revealed that the majority of amino acid residues of the α-helix are well conserved (Figs 1D and 3A and Table 3). Thus, the insertion or exchange of amino acids in the sequence of the α-helix is expected to interfere with structure and/or interactions (see above). However, right after this α-helix there is a hairpin loop between a disulphide bridge found in all TS (e.g. Cys493 and Cys503 in TconTS1a). This loop (Fig 3A) is highly flexible and does not contain

any conserved motifs or structurally relevant elements. For us it appeared a suitable target for introducing a restriction site to swap CDs and LDs from different TconTS. Indeed, the homology model of TconTS1a/TS3 showed a similar overall structure compared to TconTS1a and TconTS3 (Fig 2), including a similar size of the molecular interface (Table 1). In addition, the model also revealed an extended hydrogen bond network between CD and LD of domain swapped TconTS1a/TS3 (Tables 2 and 3). In conclusion, this model suggests that such a recombinant protein with swapped domains would fold properly and encouraged us to fuse TconTS1a-CD and TconTS3-LD as a model system *in vitro*.

We cloned and expressed recombinant TconTS1a/TS3 in eukaryotic CHO-Lec1 cells and bacteria *E. coli* Rosetta DE3 pLacI. The production of soluble active enzyme confirmed that CD and LD from different TS can be fused without complete loss of proper folding although the majority of bacterial expressed TconTS was found in the insoluble fraction. The insertion of amino acids YV (Fig 3) to facilitate domain swaps did not affect Sia transfer activity or hydrolytic release of Neu5Ac of the recombinant protein expressed in CHO-Lec1 or *E. coli* cells (Table 5). However, TconTS expressed by *E. coli* exhibit enzyme activities that were 3–4 orders of magnitude lower (S3 Fig and S2 Table) than those observed for enzymes expressed by CHO-Lec1 cells. One explanation for these observations is improper folding as indicated by the relatively high amount of insoluble TconTS. From these results it has become clear that folding of TconTS is major challenge in bacteria and probably does not assemble the protein in the native eukaryotic system. Thus CHO-Lec1 cells represent a more suitable expression system for TconTS constructs then *E. coli* in order to answer our main research questions. Therefore, we decided to exclude activity data obtained from bacterial expressed TconTS for the discussion regarding the influence of TconTS3-LD on Sia transfer efficiency of TconTS1a-CD.

One major difference between proteins expressed in bacteria and eukaryotic cells is glycosylation on asparagine residues (*N*-glycosylation), which can affect the folding process as well as the activity of the protein. For example, it has been reported that enzymatic deglycosylation of TvivTS1 expressed by *Pichia pastoris* led to a slight reduction in sialidase activity relative to the untreated TvivTS1 [51].

The presence of several putative *N*-glycosylation sites in TconTS indicates that these enzymes contain *N*-glycans in both, CD and LD [24]. Interestingly, the TconTS mAb 7/23, binds more strongly to CHO-Lec1 expressed TconTS1a compared to the enzyme when expressed by *E. coli* (Fig 5). It is unlikely that this is due to improper folding in the bacteria, since the anti-TconTS mAb 7/23 binds to the SDS-denatured protein in this experiment. However, it is likely that efficient binding of mAb 7/23 to TconTS1-LD is supported by at least one *N*-glycan. Since PNGase treatment of TconTS1a expressed by CHO-Lec1 cells strongly reduced binding of this antibody (S2 Fig). In this context it is important to note that native glycosylated TconTS was used for immunisation to generate this antibody [26]. It must be keep in mind that the eukaryotic expression system (CHO-Lec1 cells) used in this and our previous studies [14,23] leads to high-mannose type *N*-glycans [52] similar to those found on trypanosomal glycoproteins [35], leading to the assumption of a similar situation for TconTS. Along this line, it should be noted that we have demonstrated that *N*-glycosylation plays a more pivotal role in regulating TS-activity in a parallel study [53]. We found that *N*-glycans of TconTS1 enhance substrate affinity through intramolecular interactions with amino acid residues of the catalytic pocket.

## Cooperativity between CD and LD of TconTS1a

According to the assumption that the LD of TconTS indirectly influences enzyme activities, it was expected that LD of the less active TconTS3 may decrease the enzymatic activities, if

attached to the CD of the more active TconTS1a. In order to test this hypothesis and to create a proof-of-concept we cloned and expressed TconTS1a/TS3 in CHO-Lec1 cells as a domain swap model. Surprisingly, replacement of the LD in TconTS1a with TconTS3-LD does not lead to a reduction of Sia transfer activity (Table 5). This indicates that TconTS3-LD has no negative effect on trans-sialylation of TconTS1a-CD. However, hydrolytic release of Neu5Ac was reduced by 82% in case of TconTS1a/TS3 compared to TconTS1a (Table 5) in the presence of the acceptor substrate lactose resulting in a five-fold higher Sia transfer efficiency relative to that of TconTS1a. It can be concluded that TconTS3-LD suppressed hydrolytic release of Neu5Ac in TconTS1a/TconTS3 relative to TconTS1a-LD.

Previously it was demonstrated that TconTS1a-LD binds to oligomannose trisaccharides, whereas no such interaction was observed for TconTS3-LD [24]. This raised the question whether occupation of the carbohydrate binding site in TconTS1a-LD modulates the enzymatic reaction at the active site of CD of TconTS1a. Therefore, we investigated whether 1,4β-mannotriose, one of the potential binding partners of TconTS1a-LD [24], can influence the activities of TconTS1a by occupying the binding side of the LD. Our results clearly demonstrated that in the presence of 5 mM 1,4β-mannotriose Sia transfer efficiency was increased by a factor of five, due to a suppression in hydrolytic release of the reaction side product Neu5Ac (Table 5). However, direct competition of 1,4β-mannotriose for the Sia acceptor binding-site in TconTS-CD can be excluded, since TS activity is not altered (Table 5). Strikingly, this 1,4β-mannotriose-dependent increase in transfer efficiency resembles almost precisely the effect of replacing the LD of TconTS1a with TconTS3-LD. Considering the diversity of TS in African trypanosomes, it will be interesting to generate also other domain swapped TconTS constructs for a deeper understanding of the interplay between the different CD and LD of trypanosomal TS.

We propose that structural architecture and orientation of the carbohydrate binding-site in TconTS-LD provides the possibility of multivalent ligands such as neighbouring cell surface glycoconjugates to bind to the LD with corresponding effects on the supramolecular arrangement of the glycocalyx components. Furthermore, the distance between the proposed binding site in the LD and the active site of the CD could allow cooperative interactions with both CD and LD. For example, it has been reported that glutamic acid/alanine-rich protein (GARP), a *T. congolense* stage specific glycoprotein, was co-purified with TS-form 1 but not TS-form 2, both isolated from *T. congolense* procyclic cultures [26]. Interestingly, for TS-form 1 significantly higher Sia transfer activity was observed, whereas relative sialidase activity was higher in TS-form 2, although for both preparations, TS-form 1 and TS-form 2, the same donor and acceptor substrate preferences were described [26].

Furthermore, GARP is glycosylated with high-mannose type and galactosyl oligosaccharides [32] and is shown to be sialylated in TS reactions [5]. Together, these findings provide strong evidence for its multivalent binding potential to TconTS-CD and LD, as discussed earlier [24]. One example of a combination of carbohydrate binding and hydrolysing domains and such a multivalent binding scenario in one protein represent *Vibrio cholerae* neuraminidase. In addition to the catalytic domain, it consists of two lectin-like domains from which one exhibit galactose-binding but lack hydrolytic activity [20,47]. Authors have proposed that this additional cooperative binding promotes adhesion of the bacteria to epithelial cells in order to bring the neuraminidase close to its substrates present on the host cell surface, thus promoting enzymatic activity. A similar functionality is quite conceivable for TconTS and TS in general.

In summary, this study represents a proof-of-principle for the cooperative interplay of CD and LD and the influence of LD on TconTS enzymatic activities. The demonstration of this influence of TconTS3-LD on TconTS1a enzymatic activity provides novel insight into the complexity of TS catalytic mechanisms by demonstrating the modulatory effect of the LD and

its interaction with glycan structures for example present on the surface of trypanosomes or host cells.

Furthermore, results obtained from our domain swapped model TconTS1a/TS3 (Table 5) provide the proof-of-principle that TconTS3-LD modulates Sia transfer efficiency of TconTS1a-CD. This study together with our previous results [24] demonstrating the ability of TconTS-LD to bind oligomannose-oligosaccharides and our current study [53] demonstrating the involvement of oligomannose *N*-glycans of TconTS on enzymatic activity provide new aspects of biochemical properties of TconTS and TS in general. Unravelling the roles played by glycans interacting with TS-LD and its modulation of TS activity opens new perspectives, not only for a better understanding of their mechanisms, but it also provides ideas how glycosylation can modulate other systems as well.

## Supporting information

**S1 Table. List of primers used for cloning and Eco105I restriction site insertion.**
(PDF)

**S2 Table. Specific catalytic activities of different TconTS enzymes expressed by *E. coli* Rosetta pLacI.**
(PDF)

**S1 Fig. Expression of TconTS constructs in *E. coli* Rosetta pLacI. A:** Schematic presentation of recombinant TconTS construct for expression in *E.coli* Rosetta pLacI. Fusion tags flanking TconTS are: MBP: maltose binding protein tag, TEV: *tobacco etch virus* protease cleavage site, 3C: human rhinovirus 3C protease cleavage site, SNAP: SNAP-tag, *Strep*: Strep-tag. **B:** SDS-PAGE of purified TconTS constructs. 1–2 μg of protein were loaded as indicated on a 10% SDS polyacrylamide gel, which was stained with Coomassie Brilliant Blue after electrophoresis. TconTS constructs with Eco105I restriction site inserted are indicated by *. Lane 3 comprise the domain swapped construct TconTS1a/TS3.
(TIF)

**S2 Fig. PNGaseF treatment of CHO-Lec1 expressed recombinant TconTS1a.** CHO-Lec1 expressed recombinant TconTS1a was deglycosylated using PNGaseF under native or denaturing conditions (denat. cond.) as described under Methods. 1 μg of TconTS1a was used in SDS-PAGE analysis with subsequent Coomassie Brilliant Blue staining and 100 ng in Western blots using anti-TconTS mAb 7/23 and anti-*Strep*-tag for detection as indicated (details under Methods).
(TIF)

**S3 Fig. Enzymatic activities of bacterial expressed recombinant TconTS1a. A-B:** TconTS1a concentration dependent production of 3'SL (A) and hydrolytic release of Neu5Ac (B) using up to 1 μg of purified, bacterial expressed, recombinant TconTS. TS reactions were set up and analysed as described under Methods. Standard conditions with 100 μg fetuin (600 μM fetuin-bound Sia) and 2 mM lactose as Sia donor and acceptor substrates were incubated for 30 min at 37˚C. **C-D:** Time dependency of 3'SL production and hydrolytic release of Neu5Ac. Reactions were incubated for the indicated times ranging from 0–1440 min with 1 μg of purified TconTS1a with standard fetuin and lactose concentrations (see Methods). Data points are means ± standard deviation of triplicates.
(TIF)

**S1 Data. Plasmid DNA sequence: pDEF-Transin-TconTS1a(Eco105I)-SNAP-Strep**
(FASTA)

**S2 Data. Plasmid DNA sequence: pET28a-His-MBP-TconTS1a(Eco105I)-SNAP-Strep** (FASTA)

## Acknowledgments

We thank Nazila Isakovic for excellent technical assistance. We are thankful to Dr. Judith Weber for helpful discussions.

## Author Contributions

**Conceptualization:** Mario Waespy, Sørge Kelm.

**Data curation:** Mario Waespy, Thaddeus Termulun Gbem, Nilima Dinesh Kumar, Shanmugam Solaiyappan Mani, Jana Rosenau.

**Formal analysis:** Mario Waespy, Thaddeus Termulun Gbem, Nilima Dinesh Kumar, Shanmugam Solaiyappan Mani, Jana Rosenau.

**Funding acquisition:** Sørge Kelm.

**Investigation:** Mario Waespy, Thaddeus Termulun Gbem, Nilima Dinesh Kumar, Shanmugam Solaiyappan Mani, Jana Rosenau.

**Methodology:** Mario Waespy.

**Project administration:** Mario Waespy.

**Resources:** Sørge Kelm.

**Supervision:** Mario Waespy.

**Validation:** Mario Waespy, Thaddeus Termulun Gbem, Nilima Dinesh Kumar, Shanmugam Solaiyappan Mani, Jana Rosenau, Frank Dietz, Sørge Kelm.

**Visualization:** Mario Waespy.

**Writing – original draft:** Mario Waespy.

**Writing – review & editing:** Mario Waespy, Thaddeus Termulun Gbem, Nilima Dinesh Kumar, Shanmugam Solaiyappan Mani, Jana Rosenau, Frank Dietz, Sørge Kelm.

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
