## [Decision Letter · Decision Letter 0]

12 Sep 2021

Dear Dr Waespy,

Thank you very much for submitting your manuscript "Cooperativity of catalytic and lectin-like domain of T. congolense trans-sialidase modulates its catalytic activity" for consideration at PLOS Neglected Tropical Diseases. As with all papers reviewed by the journal, your manuscript was reviewed by members of the editorial board and by several independent reviewers. In light of the reviews (below this email), we would like to invite the resubmission of a significantly-revised version that takes into account the reviewers' comments. 

We cannot make any decision about publication until we have seen the revised manuscript and your response to the reviewers' comments. Your revised manuscript is also likely to be sent to reviewers for further evaluation.

Sincerely,

Carlos A. Buscaglia, PhD

Associate Editor

Alvaro Acosta-Serrano

Deputy Editor

Reviewer's Responses to Questions

**Key Review Criteria Required for Acceptance?**

**Methods**

-Are the objectives of the study clearly articulated with a clear testable hypothesis stated?

-Is the study design appropriate to address the stated objectives?

-Is the population clearly described and appropriate for the hypothesis being tested?

-Is the sample size sufficient to ensure adequate power to address the hypothesis being tested?

-Were correct statistical analysis used to support conclusions?

-Are there concerns about ethical or regulatory requirements being met?

Reviewer #1: The structure of this article is rather confusing. Authors analyze the impact of the LD domain of TconTSs on the enzymatic activity. After a detailed analysis through modelling several TSs originating from different organisms, they reach the already known conclusion that CD and LD domains are held together through non-covalent unions, which defines a large area of interaction. As they hypothesize that LD domains influence the enzymatic activity, they decide to exchange the LD domain between several different TSs. However, they only focus in one construction between two TconTSs (so, why the structural analysis of so many TSs and on all the TconTSs?)

Reviewer #2: See general comments

Reviewer #3: The objectives of the study are cleary presented and from a methodological point of view, the study is well conducted. However, the authors fall short on the sample size (no. of TS swaps) to adequately address the aims of the study: to understand the effect of the LD-domain on TS activity. Statistical analysis support their conclusions and there are no specific ethical of regulatory requirements involved.

**Results**

-Does the analysis presented match the analysis plan?

-Are the results clearly and completely presented?

-Are the figures (Tables, Images) of sufficient quality for clarity?

Reviewer #1: Authors discard the use of the alpha helix that links the CD with the LD as swapping target, as it seems structural. They instead decide to use a Cys-flanked loop known to be variable and even disposable for enzyme activity. A decision that was experimentally proven as the correct one. Authors then proceed to detail all “possible” constructs in Table 4, but make and test only one (TS1/TS3). They state (line 527) “The crucial question of this study was which impact replacing the LD of TconTS1 with that from TconTS3 would have on the enzymatic activity of the fusion protein TconTS1a/TS3”, so why mention all the variants? And why include TS2 and TS4 in Fig 4C? Why are the authors not focusing on the only experimental chimeric molecules they generated? It is very confusing. In addition, the legend corresponding to Fig 4C, mentions a non-existent lane 6.

Fig 4E shows that the glycosylation sites are conserved following the Eco105I restriction site insertion, a completely expected result as it inserts a YV dipeptide without disrupting any glycosylation consensus sequences. Besides, glycosylation is later demonstrated by PNGase treatment in Fig 5.

The major concern is the determination of the TS/sialidase activities themselves. The sialidase activity should be properly determined in the absence of an acceptor sugar as these (and other) authors have routinely done. This will measure its ability to hydrolyze the sugar, thus providing an accurate determination of the sialidase activity, which will then allow for comparative analysis with other reports. Testing free Sia with standard donor-acceptor molecules (specially under saturating conditions) does not accurately determine sialidase activity. The absence of determination of sialidase activities by MuNANA or fetuin hydrolysis in the absence of acceptor suggars, as the authors usually do, is one of the main flaws of this report. 

Line 447: …for TconTS3 no sialidase activity could be detected under conditions used [23].

However, ref 23 conditions are different as authors did not use acceptor molecules there, and thus the results obtained here are not comparable.

Line 553: ….after 120 min of incubation indicates that pTconTS1a does not release free Neu5Ac from fetuin but rather from the reaction product 3’SL.

It seems authors consider the Sia transfer occurs in strictly unidirectionaly from fetuin to lactose, probably due to the starting concentrations (600uM vs. 2mM). However, once 3’SL is generated, it becomes a Sia donor not only in hydrolysis but also transferences and can even back regenerate fetuin. Is there any known substrate preference in these enzymes?

Line 554: "Along this line, when considering the TS over sialidase activity (TS/sialidase) ratio pTconTS1a shows clear preference towards Sia-transfer to lactose (Table 5)."

Sialidase activity measured as free Sia in the presence of acceptor sugars is not comparable with previous references (see above).

In Table 5 authors seem to use Neu5Ac as equivalent to Sia. However, in addition to Neu5Ac, bovine fetuin also contains glycoyl sialic. Does these enzymes show any substrate preference?

Reviewer #2: See general comments

Reviewer #3: Trans-sialidase (TS) is an important enzyme produced by Trypanosomatids. As pointed out by the authors, due to their exclusive activity in these organisms, these enzymes are a pharmacological target of interest for the development of anti-trypanosmatid agents. 

TS enzymes contain two domains, the sialidase catalytic domain (SD) and the LamG-like lectin-like domain (LD). Here, the authors performed extensive bioinformatic/structural studies to understand the interaction between the domains and how they could affect the overall activity of the enzyme. They measure this as the ratio of TS activity x sialidase activity, or failure to transfer the sialic acid to a suitable acceptor molecule. 

However, although they have presented some very interesting structural data, it seems to me that the study in its present form is still preliminary and lacks more in-depth analysis and experiments. Most importantly, their overall conclusion is based on a single fusion construct with contrasting data (bacterial x mammalian). 

Strong points: the authors have performed a very nice study to map the contact points between the SD and LD domain. 

Week points: based on their map, they chose to swap domains of two TS to generate a single swap construct (TconTS1a/TS3) which was produce in bacterial and mammalian cells. In bacteria, this swapped construct changed its activity profile from a more TS to a more SA enzyme activity. Nevertheless, the same construct when produce in mammalian cells display the opposite effect, with even more TS activity.

It seems to me that it is difficult to reach any conclusion regarding the effect of domain combination in the TS family based on these results. The author would have to generate many more constructs (SD1/LD2 x SD2/LD1 compared their corresponding wt1 and wt2, etc... for at least 2 or 3 different TS enzymes). 

The way I see this is: trypanosomatids contain hundreds of copies of TS genes that were generated by gene duplication during evolution. Nature probably selected combinations of SD/LD copies that worked well and eliminated those that worked poorly (natural swapped generated during evolution). So, it is not surprising that changes of the LD domain of a given TS gene would change the behavior of the enzyme. The question is: what can we learn from these different combination? What is their molecular base? Based on this study in its present form, not much, unfortunately. 

Minor issues:

Table 4 seems dispensable. The authors showed all possible combinations and then generated a single one? 

Furthermore, the results regarding PNGaseF digestion to remove N-linked glycosylation is strange and it seems like a side project. Usually, N-glycosylation usually blocks access to antibodies (cryptic epitopes). In these cases, it seems to me that lack of glycosylation may simply affect domain folding or the display of the epitope on the nitrocellulose/PVDF membrane; a technical artifact. Further experiments would be necessary to address this conundrum.

Other authors have showed the the LamG-lectin like domain bind to other substrates as well (ie., cytokeratin - Teixeira et al., PLOS Negl Trop Dis, 2015). None of these studies were cited by the authors or discussed in terms of how it could affect TS activity (by bringing specific acceptor molecules close to the SD domain?).

**Conclusions**

-Are the conclusions supported by the data presented?

-Are the limitations of analysis clearly described?

-Do the authors discuss how these data can be helpful to advance our understanding of the topic under study?

-Is public health relevance addressed?

Reviewer #1: Line 601: "The presence of 1,4β-mannotriose did not have a significant effect on the production of 3’SL (Table 5). Interestingly, in the presence of 1,4β-mannotriose the release of free Neu5Ac was about 6-fold lower."

This is an interesting conclusion but, again, it should be determined in the absence of acceptor sugars.

Reviewer #2: See general comments

Reviewer #3: Although the authors performed a very interesting and promising study regarding the structural requirements for the interaction between the SD and LD domains of TS enzymes, they fall short on the number of swap combinations they performed in order to reach any meaningful conclusion.

**Editorial and Data Presentation Modifications?**

Reviewer #1: Please use Trypanosoma as genus in the title instead of T. 

Line 30: the activated Sia used…. Please expand Sia here (first use)

Line 31: been done on enzymatic activities of TS from T. cruzi… use Trypanosoma (first use) same in the lines 33 and 34 (even for different species).

Line 40: acid (Sia) transfer activities … abbreviate to Sia from line 30 on.

Line 65 replace Trypansoma for Trypanosoma.

Line 84 characterised sialidase from M. viridifaciens… expand the genus.

Line 101 replace Tryoanosoma by Trypanosoma 

Line 155: insert "lactose" to not confuse readers.

Line 321 Marcobdella decora please check the genus (Macrobdella).

Lines 374 and 375: correct spelling of “replayced”

Line 481-482: "The resulting question was whether the binding epitope of the antibody is located within the CD or LD."

This was solved in previous reports from this lab, where they state: “Further experiments provided evidence that the epitope is located in the LDs” (Ref 23).

As this article should focus on the activity of the swapped enzymes (as authors state in lines 527-529), the mapping of this epitope is not central to it and thus should be reduced to the useful application of the mAb to determining the LD, as pointed out in Fig 4D. Authors should also emphasize that, as the mAb recognized the p-enzymes in Fig 5C, the epitope partially involves the glycosylation of the enzyme (referring to Fig 5 as a supplementary figure). These modifications would reduce text and avoid distracting the reader.

Several parts of the discussion only reiterate comments already made in Results and thus should be reduced to avoid redundancy. (Discussion of the contact areas among protein domains, selection of the loop for swapping, etc.)

Reviewer #2: See general comments

Reviewer #3: (No Response)

**Summary and General Comments**

Reviewer #1: The article is interesting to clarify the biochemical properties of the TconTSs. However, it seems not structurally well written making it very confusing. Readers expect more results early on when they analyze many different TSs and then the TconTS between them, but end up studying a single chimeric construct. I can accept this as a "proof of principle" that the LDs actually play a significant role on the enzymatic activity. However, the sialidase activity will be more accurately evaluated in the absence of acceptor sugars.

Reviewer #2: In this manuscript the authors expand their previous results that showed that lectin domain (LD) of Trypanosoma trans-sialidase (TS) binds to oligosaccharides and now suggested that this binding modulates the transglycosylation reaction reducing the transference to water. Such conclusion could suggest modulatory effects of the enzyme by its own glycosylation or when it recognize different substrates. To show this modulation, the authors initially used TS of Trypanosoma congolense and generate structural models which indicate possible interactions through hydrogen bounds between the catalytic domain (CD) and the LD. The models also allowed them to swap the LD domains between a T. congolense TS1 and TS3, this later showing less trans activity, and verify the influence of the LD on both reactions. Finally, they showed that addition of mannotriose to recombinant enzymes decrease the transfer to water. 

The results add some novel information about this possible regulatory mechanism and indicate that the LD domain is key for preventing the hydrolysis reactions. However, the manuscript is quite difficult to follow as it mix key findings with controls and system definitions. There are some issues that confounds the reader and require a better organization of the text. I will list some topics that could be modified and improved. 

Major points

1. The use of recombinants obtained in E. coli versus mammalian cells create several doubts and it is difficult to distinguish whether the differences are due to protein glycosylation and improper folding and possible aggregation and/or denaturation. I think the use of the two systems does not help to explain a role of the LD domain in the activity modulation as it could just help to fold the beta-barrel. It would be interesting to further characterize the aggregation state, alpha-helix content or other structural property in parallel.

2. The results of the mAb specific for TScon1, which recognize only after glycosylation is quite complex. As ut recognize the proteins after SDS-PAGE, it seems that it is not directed to conformational epitopes. Therefore, it does not help to reach the main conclusion and confounds the reader. 

3. It is unclear whether the glycosylation type (high mannose and complex) is related to the activity and if they act by intramolecular interactions. This also confounds the reader to understand the role of glycosylation versus the addition of exogenous oligosaccharides. 

4. The strict comparison of the TS versus hydrolysis activity could be better performed (see below).

Specific points

-Line 39: It is unclear what is TS3 in the abstract for those who are not familiar with TScon

-Line 42: It is unclear the relation between the mannotriose and the swap experiment and the conclusion have no clear support on the previous sentence.

-The last paragraph of the Introduction describes the results in an unusual way. I would suggest modifying it.

-Line 168. Fig. 5 citation appears before other figures (check the order in the final version)

-It would be interesting to have the plasmid sequences deposited or available.

-Line 223. Fig. 4A citation appears before other figures (check the order in the final version)

-Line 224, CHO are derived from ovary. I’m not sure if are typical fibroblasts.

-Line 436. The sentence appears unlinked the description of expression in bacteria. This part of the text should be reorganized.

-Line 460, it should be stated that p is prokaryotic and e eukaryotic in the legend. 

-Line 524, it would be interesting to indicate the putative N-glycosylation sites in the protein.

-Line 541, the legend state that 3 ug of enzyme and in the figure is annotated 1 ug. 

-Line 552, the delayed release of free Neu5Ac could be due to a secondary release from SL formed after transfer from fetuin to lactose as stated. However, the results are still insufficient to proof this. A better kinetic analysis could be performed.

-Table 5. The time used for measurements should be stated. Are these initial velocities? The results are clearly different but to make statements and compare the ratio of TS/sialidase, it would be important to perform Michaelis determination (Km and Vms). 

- Table 5. The difference in activity could be a consequence of proper folding due to glycosylation, membrane export, cysteine bridge formation and glycosylation. But part of the prokaryotic enzyme could be denatured or aggregated. Further purification could be necessary. This could reflect the low production of protein in E. coli and their low specific activity when compared to the enzymes produced in CHO cells. I noticed that this is discussed but to be published some additional evidences should be provided. Also, it is important to describe how the proteins are stored. 

- Line 600. Is 5 mM of mannotriose compatible with its affinity. Perhaps, it would be important to titrate the effect to measure binding.

Reviewer #3: (No Response)

PLOS authors have the option to publish the peer review history of their article (what does this mean?). If published, this will include your full peer review and any attached files.

Reviewer #1: No

Reviewer #2: No

Reviewer #3: No
---

## [Decision Letter · Decision Letter 1]

23 Jan 2022

Dear Dr Waespy,

We are pleased to inform you that your manuscript 'Cooperativity of catalytic and lectin-like domain of T. congolense trans-sialidase modulates its catalytic activity' has been provisionally accepted for publication in PLOS Neglected Tropical Diseases.

Before your manuscript can be formally accepted you will need to complete some formatting changes, which you will receive in a follow up email. Please use this instance to address the minor modifications (typos and grammatical mistakes) raised by reviewers #1 and #2. A member of our team will be in touch with a set of requests.

Best regards,

Carlos A. Buscaglia, PhD

Associate Editor

Alvaro Acosta-Serrano

Deputy Editor

Reviewer's Responses to Questions

**Key Review Criteria Required for Acceptance?**

**Methods**

-Are the objectives of the study clearly articulated with a clear testable hypothesis stated?

-Is the study design appropriate to address the stated objectives?

-Is the population clearly described and appropriate for the hypothesis being tested?

-Is the sample size sufficient to ensure adequate power to address the hypothesis being tested?

-Were correct statistical analysis used to support conclusions?

-Are there concerns about ethical or regulatory requirements being met?

Reviewer #1: (No Response)

Reviewer #2: All adequate

Reviewer #3: ...

**Results**

-Does the analysis presented match the analysis plan?

-Are the results clearly and completely presented?

-Are the figures (Tables, Images) of sufficient quality for clarity?

Reviewer #1: (No Response)

Reviewer #2: The answers and modifications introduced made it very clear.

Reviewer #3: ...

**Conclusions**

-Are the conclusions supported by the data presented?

-Are the limitations of analysis clearly described?

-Do the authors discuss how these data can be helpful to advance our understanding of the topic under study?

-Is public health relevance addressed?

Reviewer #1: (No Response)

Reviewer #2: Are supported and clear.

Reviewer #3: ...

**Editorial and Data Presentation Modifications?**

Reviewer #1: Authors have clearly improved the presentation of the research aims and data.

Some Minor concerns remain:

Please use Trypanosoma as genus in the title and in the abstract instead of just T.

Line 89: characterised sialidase from M. viridifaciens… please expand the genus.

Line 32: been done on enzymatic activities of TS from T. cruzi… use Trypanosoma (first use) same in the lines 34 and 35 (even for different species).

Line 70 replace Trypansoma for Trypanosoma.

Line 106: replace Tryoanosoma by Trypanosoma

Line 323 Marcobdella decora please check the genus (Macrobdella).

Line 378: correct spelling of “replayced”

Line 146: delete the ? sign

Line 147 … the presence of an suitable …. (use "a" instead)

Lines 408 to 412 where authors discuss the absence of homology in the four TconTSs after the conserved Cys bridge, it might be of interest to note that in the TcruTS this loop plays no structural role but constitutes a relevant antigenic region that allows many different sequences to be located there (PMID: 12134236).

Reviewer #2: Very good.

Reviewer #3: ...

**Summary and General Comments**

Reviewer #1: (No Response)

Reviewer #2: Small corrections to be made on proof corrections.

Lne 70: Trypansoma

Line 378, relayced

Line 452, consructs

Line 481, an C-terminal

Line 490, instead of is located in, just in

Line 532 E.coli

Line 673, Therefore,

Line 730, For example,

Reviewer #3: The authors have addressed most of my previous queries. Although it would be interesting to see the effect of further swaps, nevertheless, the results do indicate that the LamG-like domain somehow direct the TS catalytic domain to optimize sialidase transfer activity.

PLOS authors have the option to publish the peer review history of their article (what does this mean?). If published, this will include your full peer review and any attached files.

Reviewer #1: No

Reviewer #2: **Yes: **Sergio Schenkman

Reviewer #3: No

---

## [Editor Report · Acceptance letter]

2 Feb 2022

Dear Dr. Waespy,

We are delighted to inform you that your manuscript, "Cooperativity of catalytic and lectin-like domain of </i>Trypanosoma congolense</i> trans-sialidase modulates its catalytic activity," has been formally accepted for publication in PLOS Neglected Tropical Diseases.

Best regards,

Shaden Kamhawi

co-Editor-in-Chief

Paul Brindley

co-Editor-in-Chief
